# Fine-Tuning Approach for Segmentation of Gliomas in Brain Magnetic Resonance Images with a Machine Learning Method to Normalize Image Differences among Facilities

**DOI:** 10.3390/cancers13061415

**Published:** 2021-03-19

**Authors:** Satoshi Takahashi, Masamichi Takahashi, Manabu Kinoshita, Mototaka Miyake, Risa Kawaguchi, Naoki Shinojima, Akitake Mukasa, Kuniaki Saito, Motoo Nagane, Ryohei Otani, Fumi Higuchi, Shota Tanaka, Nobuhiro Hata, Kaoru Tamura, Kensuke Tateishi, Ryo Nishikawa, Hideyuki Arita, Masahiro Nonaka, Takehiro Uda, Junya Fukai, Yoshiko Okita, Naohiro Tsuyuguchi, Yonehiro Kanemura, Kazuma Kobayashi, Jun Sese, Koichi Ichimura, Yoshitaka Narita, Ryuji Hamamoto

**Affiliations:** 1Division of Molecular Modification and Cancer Biology, National Cancer Center Research Institute, 5-1-1 Tsukiji, Chuo-ku, Tokyo 104-0045, Japan; satoshi.takahashi.fy@riken.jp (S.T.); kazumkob@ncc.go.jp (K.K.); rhamamot@ncc.go.jp (R.H.); 2Cancer Translational Research Team, RIKEN Center for Advanced Intelligence Project, 1-4-1 Nihonbashi, Chuo-ku, Tokyo 103-0027, Japan; 3Department of Neurosurgery and Neuro-Oncology, National Cancer Center Hospital, 5-1-1 Tsukiji, Chuo-ku, Tokyo 104-0045, Japan; yonarita@ncc.go.jp; 4Division of Brain Tumor Translational Research, National Cancer Center Research Institute, 5-1-1 Tsukiji, Chuo-ku, Tokyo 104-0045, Japan; kichimur@ncc.go.jp; 5Department of Neurosurgery, Osaka University Graduate School of Medicine, 2-2 Yamadaoka, Suita, Osaka 565-0871, Japan; mail@manabukinoshita.com (M.K.); h-arita@nsurg.med.osaka-u.ac.jp (H.A.); 6Department of Diagnostic Radiology, National Cancer Center Hospital, 5-1-1 Tsukiji, Chuo-ku, Tokyo 104-0045, Japan; mmiyake@ncc.go.jp; 7Artificial Intelligence Research Center, National Institute of Advanced Industrial Science and Technology, 2-3-26, Aomi, Koto-ku, Tokyo 135-0064, Japan; rkawaguc@cshl.edu (R.K.); sesejun@gmail.com (J.S.); 8Department of Neurosurgery, Graduate School of Medical Sciences, Kumamoto University, 1-1-1 Honjo, Chuo-ku, Kumamoto 860-8556, Japan; nshinojima@kuh.kumamoto-u.ac.jp (N.S.); mukasa@kumamoto-u.ac.jp (A.M.); 9Department of Neurosurgery, Kyorin University Faculty of Medicine, 6-20-2, Sinkawa, Mitaka, Tokyo 181-8611, Japan; kusaito-tky@umin.ac.jp (K.S.); mnagane@ks.kyorin-u.ac.jp (M.N.); 10Department of Neurosurgery, Dokkyo Medical University, 880 Kitakobayashi, Mibu, Shimotsugagun, Tochigi 321-0293, Japan; ryouhei-ohtani@umin.ac.jp (R.O.); fhiguchi@dokkyomed.ac.jp (F.H.); 11Department of Neurosurgery, Tokyo Metropolitan Komagome Hospital, 3-18-22 Honkomagome, Bunkyo-ku, Tokyo 113-8677, Japan; 12Department of Neurosurgery, Faculty of Medicine, The University of Tokyo, 7-3-1 Hongo Bunkyo-ku, Tokyo 113-8655, Japan; tanakas-tky@umin.ac.jp; 13Department of Neurosurgery, Graduate School of Medical Sciences, Kyushu University, 3-1-1 Maidashi, Higashi-ku, Fukuoka 812-8582, Japan; hatanobu@ns.med.kyushu-u.ac.jp; 14Department of Neurosurgery, Tokyo Medical and Dental University, 1-5-45 Yushima, Bunkyo-ku, Tokyo 113-8510, Japan; tamura.nsrg@tmd.ac.jp; 15Department of Neurosurgery, Graduate School of Medicine, Yokohama City University, 3-9 Fukuura, Kanazawa-ku, Yokohama, Kanagawa 236-0004, Japan; ktate12@yokohama-cu.ac.jp; 16Department of Neuro-Oncology/Neurosurgery, Saitama Medical University International Medical Center, 397-1 Yamane, Hidaka, Saitama 350-1298, Japan; rnishika@saitama-med.ac.jp; 17Department of Neurosurgery, Kansai Medical University, 2-5-1 Shinmachi, Hirakata, Osaka 573-1010, Japan; nonakamasa65@gmail.com; 18Department of Neurosurgery, National Hospital Organization Osaka National Hospital, 2-1-14 Hoenzaka, Chuo-ku, Osaka 540-0006, Japan; yokita4246@gmail.com (Y.O.); yonehirok@gmail.com (Y.K.); 19Department of Neurosurgery, Osaka City University Graduate School of Medicine, 1-4-3 Asahi-machi, Abeno-ku, Osaka 545-8585, Japan; uda@med.osaka-cu.ac.jp (T.U.); ntsuyuguchi@gmail.com (N.T.); 20Department of Neurological Surgery, Wakayama Medical University School of Medicine Wakayama, 811-1 Kimiidera, Wakayama 641-8509, Japan; junfukai@wakayama-med.ac.jp; 21Department of Neurosurgery, Osaka International Cancer Institute, 3-1-69 Ootemae, Chuo-ku, Osaka 541-8567, Japan; 22Department of Neurosurgery, Kindai University Faculty of Medicine, 377-2 Ohnohigashi, Osaka-Sayama, Osaka 589-8511, Japan; 23Department of Biomedical Research and Innovation, Institute for Clinical Research, National Hospital Organization Osaka National Hospital, 2-1-14 Hoenzaka, Chuo-ku, Osaka 540-0006, Japan; 24Humanome Lab, 2-4-10 Tsukiji, Chuo-ku, Tokyo 104-0045, Japan

**Keywords:** glioma, machine learning, MR images, fine-tuning, deep learning

## Abstract

**Simple Summary:**

This study evaluates the performance degradation of machine learning models for segmenting gliomas in brain magnetic resonance images caused by domain shift and proposed possible solutions. Although machine learning models exhibit significant potential for clinical applications, performance degradation in different cohorts is a problem that must be solved. In this study, we identify the impact of the performance degradation of machine learning models to be significant enough to render clinical applications difficult. This demonstrates that it can be improved by fine-tuning methods with a small number of cases from each facility, although the data obtained appeared to be biased. Our method creates a facility-specific machine learning model from a small real-world dataset and public dataset; therefore, our fine-tuning method could be a practical solution in situations where only a small dataset is available.

**Abstract:**

Machine learning models for automated magnetic resonance image segmentation may be useful in aiding glioma detection. However, the image differences among facilities cause performance degradation and impede detection. This study proposes a method to solve this issue. We used the data from the Multimodal Brain Tumor Image Segmentation Benchmark (BraTS) and the Japanese cohort (JC) datasets. Three models for tumor segmentation are developed. In our methodology, the BraTS and JC models are trained on the BraTS and JC datasets, respectively, whereas the fine-tuning models are developed from the BraTS model and fine-tuned using the JC dataset. Our results show that the Dice coefficient score of the JC model for the test portion of the JC dataset was 0.779 ± 0.137, whereas that of the BraTS model was lower (0.717 ± 0.207). The mean Dice coefficient score of the fine-tuning model was 0.769 ± 0.138. There was a significant difference between the BraTS and JC models (*p* < 0.0001) and the BraTS and fine-tuning models (*p* = 0.002); however, no significant difference between the JC and fine-tuning models (*p* = 0.673). As our fine-tuning method requires fewer than 20 cases, this method is useful even in a facility where the number of glioma cases is small.

## 1. Introduction

Glioma is the most frequent primary intracerebral tumor in adults. The prognosis for most glioma patients is fatal, and the disease has a significant impact on the physical, psychological, and social status of the patients and their families [1]. The World Health Organization (WHO) grade of glioma represents its biological behavior and correlates with the number of accumulated gene mutations [2,3]. Although the treatment strategy differs with the type of glioma and WHO grade, a combination of surgical resection, radiotherapy, and chemotherapy is generally applied [4,5,6,7]. Because of the invasive nature of gliomas, it is difficult to remove all tumor cells within an adequate safety margin; however, the prognosis of patients with glioma improves as the extent of resection increases [8]. Therefore, it is valuable to identify tumor boundaries before surgery.

Magnetic resonance (MR) imaging is routinely used to diagnose and determine treatment strategies for gliomas. Compared with computed tomography, MR imaging has the advantage of providing more detailed information regarding tumors and surrounding tissues. Therefore, MR imaging is preferably used to (1) determine the extent of a tumor by neuroradiologists, (2) monitor the boundary of lesions when neurosurgeons perform glioma resection with a navigation system, and (3) evaluate the effect of treatment by neurooncologists. However, manual segmentation of brain tumors from numerous MR images generated in clinical routine is a time-consuming and challenging task. Hence, there is a need for a technology that allows automated brain tumor segmentation. It is an urgent mission to build a machine learning model for the segmentation of brain tumors.

The most notable network architecture in machine learning for segmenting tumors is U-Net [9], and several improved models have already been developed [10,11,12,13,14,15]. For example, Myronenko won the first prize in the Brain Tumor Segmentation Competition (BraTs) in 2018, with a very high Dice coefficient score (≥ 0.90) [16,17].

Despite the current rapid progress in using machine learning and deep learning technologies in the medical field [18,19,20,21,22,23,24,25,26], the clinical application of machine learning models for tumor segmentation still requires a significant amount of progress. One reason for the delay in its clinical application appears to be the performance degradation caused by domain shift [27]. Most machine learning models for the segmentation of MR images in recent years use deep learning techniques. The aforementioned U-Net is also an example of deep learning architecture. When training these architectures, one often assumes that the training data and test data are sampled from the same distribution [28]. In medical imaging, images obtained from one facility (one imaging device and one image protocol) can be assumed as a part of one domain. Therefore, when a deep learning model is created from data acquired from facility A (domain A), the model can display high prediction accuracy for newly acquired data from domain A. However, suppose the model is applied to data from another facility, B (domain B). In that case, the performance may drop significantly if the data distributions of domains A and B are different (this is called ‘domain shift’). Thus, the clinical application of deep learning models for segmenting MR images has not been established. The performance degradation, due to the domain shift, is a critical issue that must be handled in a real clinical setting because the impact and frequency of the performance degradation caused by domain shift are unpredictable. To understand the mechanisms involved in performance degradation, due to the domain shift, we must carefully analyze data from multiple facilities and multiple domains.

In this study, we aimed (1) to clarify the detailed mechanisms of performance degradation caused by a domain shift by collecting images from 10 different facilities in Japan, and (2) to develop a method to solve this issue.

## 2. Materials and Methods

### 2.1. Ethics

This study was approved by the Ethics Committees of all the facilities that provided the images (IRB number: 2013-042) the Ethics Committee of the National Cancer Center, Tokyo, Japan (approval ID: 2016-496). All methods were performed in accordance with the Ethical Guidelines for Medical and Health Research Involving Human Subjects.

### 2.2. JC Dataset

Preoperative (before surgical resection or biopsy) MR images were tried to be collected from 951 adult diffuse glioma patients from 10 facilities in Japan between November 1991 and October 2015; the patients had participated in our previous study [29] in which we demonstrated that the combination of IDH mutation status, TERT promoter mutation status, and MGMT (O-6-methylguanine-DNA methyltransferase) methylation status refined the classification of WHO grade II-IV gliomas. Of the 951 cases, 673 cases had preoperative digital MR images in at least one sequence, and 544 cases met our criteria described below. Clinical information was collected in detail, and sequencing analysis was performed on the tumor specimens collected from these patients. We named these images the Japanese cohort (JC) dataset. The images satisfying the following three categories were selected:All four types of images, T1-weighted images (T1), T2-weighted images (T2), fluid-attenuated inversion recovery (FLAIR), and T1-weighted images with gadolinium enhancement (GdT1), were eligible.Surgical removal or biopsy was performed.Diagnostic tests, including genetic analysis of key biomarkers (IDH mutation and 1p19q), were performed following the WHO 2007 or 2016 classifications of Central Nerves System tumors.

We further divided 544 subjects into three categories, as described above. The number of patients included in the JC dataset was 1.6-times larger than that in the Multimodal Brain Tumor Image Segmentation Benchmark (BraTS) dataset. To the best of our knowledge, this is the largest glioma image dataset in which genetic/epigenetic profiles and clinical information are accessible. Table 1 presents detailed information on the clinical characteristics of the JC dataset.

### 2.3. BraTS Dataset

We also used the BraTS 2019 dataset as the BraTS dataset, including T1, T2, FLAIR, and GdT1 images of 259 high-grade glioma cases and 76 low-grade glioma cases with tumor volume of interest (VOI) information [16,30,31]. Further details regarding the BraTS dataset can be found elsewhere [16,30].

### 2.4. Creating VOI

All VOIs in our JC dataset were manually created by skilled neuroradiologists using in-house tools. As shown in Appendix A, the VOIs were created based on the information regarding four sequences. The VOI was defined as an area that may have a tumor, including edema. We included the T2 hyper lesions in the VOI because they may contain non-enhancing tumor lesions. We think that the advantages of including non-enhancing tumor outweigh the disadvantages of including ‘pure edema’ that may make noise. The created VOI included parts referred to as ‘necrotic tumor core’.

In contrast, the VOI in the BraTS datasets was classified into three parts that included ‘edema’, ‘tumor core’, and ‘necrotic tumor core’. Edema, tumor core, and necrotic tumor core were merged to transfer the information and optimize the VOI of the JC dataset, and was, therefore, redefined as the new VOI of the BraTS dataset.

### 2.5. Image Preparation

The images from the BraTS dataset were cropped and resized to 176 × 192 × 160, using a script written in Python 3.7.7 (https://www.python.org, accessed on 14 October 2020). The images from the JC dataset were skull stripped by BET [32]. Then, the skull stripped images were cropped and resized to 176 × 192 × 160 using our Python script.

### 2.6. Machine Learning Model

First, the JC dataset was randomly divided into two halves; one was used for training and fine-tuning (‘JC dataset pre-training’; Figure 1), while the other was an independent test dataset. To optimize the distribution of the cases, we ensured that both training and test datasets have approximately the same number of images from each facility. Second, three types of machine learning models were created for segmentation. Finally, the performances of these models on the independent test dataset were evaluated using the Dice coefficient score. Figure 1 shows the overall flow.

BraTS model: The BraTS dataset was split into training, validation, and test sets (60% training, 20% validation, and 20% test), and the model was trained for tumor segmentation on the test set.JC model: The pre-training part of the JC dataset was further split into training and validation sets (75% training and 25% validation), and the model was trained on the training portion of the JC dataset.Fine-tuning model: The BraTS model was fine-tuned to perform an optimized analysis in each facility. A maximum of 20 cases (randomly selected) from the pre-training portion of the JC dataset were used for fine-tuning. Here, if a facility had fewer than 20 training cases, all the training portions in the JC dataset from the facility were used for fine-tuning.

The architecture of our machine learning model (Appendix A) is based on the method described elsewhere [9,33,34]. As shown in Appendix A, we grouped all layers between two element-wise sums or merged operation layers into one block for further technical convenience. Most blocks contain three 3D convolution layers and one dropout layer. The concept of each group corresponds to the blue shaded areas in Appendix A. Each block was named in order from the shallowest to the deepest block as ‘1st down’, ’2nd down’,… ‘5th down’ and then from the deepest to the shallowest block as ’1st up’, ’2nd up’,… ’5th up’. The optimizer was set to AdaGrad, and the learning rate was set to 0.005. The deep learning models were developed using the Keras package in Python (https://keras.io, accessed on 14 October 2020). The network architectures for all three models were identical. RMSprop was used as the optimizer, and the learning rate was set to 0.0005. All the images were cropped and input to the model for segmentation as a 4ch (T1, T2, FLAIR, GdT1) 3D volume of 4 × 176 × 192 × 160.

### 2.7. Finding the Best Fine-Tuning Method

The fine-tuning model is specialized for each facility; therefore, the number of the fine-tuning model is the same as that of the facilities. To determine the most appropriate fine-tuning method, we decided to use the training data of the JC dataset from facility A because facility A had a relatively large dataset (157 cases). Twenty cases were randomly selected from the training data. The remaining training data were used for performance evaluation.

Initially, we examined three different methods: (1) The first method was to train a model in which all layers were layers with learnable parameters (learnable layers; ’fine_all’ model); (2) the second method was to train a model in which only the down path was a learnable layer (’down model’; in other words, a model in which the up path was frozen); and (3) the third method was to train a model in which only the up path was capable of learning (‘up model’; Appendix A).

Next, we investigated the effect of horizontal connection instead of the vertical connections of 3D U-Net on fine-tuning (Appendix A). In this experiment, we re-examined three different methods: (1) The first method was to train a model in which all layers were learnable layers as mentioned above (‘fine_all’ model), (2) the second method was to train a model in which learnable layers were those in the 1st down and 1st up blocks (‘down1_up1 model’), and (3) the third method was to train a model in which learnable layers were those in the 1st down, 2nd down, 1st up, and 2nd up blocks (‘down2_up2 model’). The fine-tuning method, referred to as ‘down2_up2 method’, was set as follows: (1) The optimizer was set to AdaGrad, and the learning rate was set to 0.005; (2) the learnable layers were those in 1st down, 2nd down, 1st up, and 2nd up blocks (down2_up2 model); and (3) the number of epochs was 20.

### 2.8. Overall Workflow

Figure 1 shows the overall workflow. First, we randomly divided the JC dataset into two halves; one was used for training and fine-tuning (JC pre-training dataset), whereas the other was used as an independent test dataset (JC independent test dataset). We balanced both training and test datasets according to the number of images obtained from each facility to optimize data distribution. Next, we created three types of machine learning models for segmentation. We created the BraTS model by training the algorithm using the BraTS dataset and the JC model using the JC pre-training dataset. We further established several fine-tuning models by fine-tuning the BraTS model using fewer than 20 cases from each facility among the JC pre-training dataset (Appendix A). The fine-tuning models were the BraTS models adjusted explicitly for each facility using only a limited amount of additional data. The number of the fine-tuning models was equal to the number of facilities. Finally, we evaluated each model’s performance using the Dice coefficient score (see Materials and Methods Section 2.9) against our JC independent test dataset.

### 2.9. Performance Evaluation of Segmentation Models

The performance of the segmentation models was evaluated by Dice coefficient scores. The Dice coefficient is a score that indicates the similarity between two samples. In the case of two images, the Dice coefficient score was calculated by dividing the number of pixels in the overlapping area times two by the number of pixels in both images. Therefore, the Dice coefficient score ranges from zero to one, and is one of the two images are an exact match.

### 2.10. Statistical Analysis

To randomly split the dataset, we used a random function from NumPy library (https://docs.scipy.org/doc/numpy-1.14.0/reference/routines.random.html, accessed on 14 October 2019). To get robust results, we selected Welch’s ANOVA to analyze variance and Games-Howell Post-Hoc Test as a post-hoc test. Welch’s ANOVA and Games-Howell Post-Hoc Test do not assume equal variance and sample size [35,36]. We used oneway.test function from R core package (https://www.rdocumentation.org/packages/stats/versions/3.6.2/topics/oneway.test, accessed on 21 January 2020) and one-way function with Games-Howell option from R userfriendlyscience package (https://www.rdocumentation.org/packages/userfriendlyscience/versions/0.7.2/topics/oneway, accessed on 21 January 2020). With regard to the correlation analysis, we used the corrcoef function NumPy library for Python to compute the Pearson’s correlation coefficient (https://numpy.org/doc/stable/reference/generated/numpy.corrcoef.html, accessed on 14 October 2019).

## 3. Results

### 3.1. Performance of Individual Models

#### 3.1.1. BraTS Model

The Dice coefficient score of the BraTS model on the BraTS testing dataset was 0.873 ± 0.098 (Appendix A); however, the score significantly decreased to 0.717 ± 0.207 when the BraTS model was applied to the JC independent test dataset. The results are shown in detail in Appendix A. We observed the highest Dice coefficient score in the dataset from facility J (0.817 ± 0.065) and the lowest Dice coefficient score in the dataset from facility G (0.480 ± 0.286).

#### 3.1.2. JC Model

Next, we built and trained a *JC* model. The *JC* model’s Dice coefficient score for the testing data of the JC dataset was 0.779 ± 0.137. As shown in Appendix A, we observed the highest Dice coefficient score in facility H (0.836 ± 0.051) and the lowest Dice coefficient score in facility D (0.673 ± 0.273). The fluctuation in the Dice coefficient scores among the JC model facilities was observed to be smaller than that in the BraTS model (Appendix A).

#### 3.1.3. Fine-Tuning Models

In fine-tuning models, the BraTS models were fine-tuned using a small set of data from the JC pre-training dataset. First, we searched for the best fine-tuning method using the JC pre-training dataset from facility A. Initially, we examined three different methods (Appendix A). As shown in Appendix A, the ‘fine all’ model demonstrated the best performance, leading to the conclusion that both up path training and down path training were necessary to achieve optimal performance for fine-tuning.

Next, we investigated the effect of horizontal connection instead of the vertical connections of 3D U-Net on fine-tuning (Appendix A). In this experiment, we re-examined three different methods (see method). As shown in Appendix A, ‘down2_up2 model’ and ‘fine_all model’ showed comparable performances. In general, the fewer the parameters to be tuned, the lower the computational cost and the lesser the overfitting for the target domains with small data [37]; hence, ‘down2_up2 model’ was considered to be the most efficient fine-tuning model.

#### 3.1.4. The Result of Fine-Tuning Models

Using the down2_up2 method, we fine-tuned 10 BraTS models on the JC pre-training dataset from each facility. The mean Dice coefficient score of the fine-tuning models against each facility’s JC pre-training dataset was 0.769 ± 0.138 (Appendix A).

### 3.2. Comparison of the Three Models

As shown in Figure 2, the Dice coefficient score of the fine-tuning model improved significantly compared with that of the BraTS model, and it was comparable to that of the JC model. In facilities B and J, the fine-tuning model’s Dice coefficient score was higher than that of the JC model (Appendix A). Welch’s ANOVA showed significant differences among the three groups (*p* = 0.0002). The Games-Howell Post-Hoc Test showed that there was a significant difference between the BraTS and JC models (*p* < 0.0001) and between the BraTS and fine-tuning models (*p* = 0.002) in terms of the Dice coefficient score. In contrast, there was no significant difference between the JC and fine-tuning models (*p* = 0.673).

Then, we focused on pathological diagnosis and comparison of three models. We show the results in Figure 3 and Appendix A. The Dice coefficient scores of the JC model and fine-tuning models tended to be better than those of the BraTS model, especially for oligodendroglioma and glioblastoma.

### 3.3. Comparison of the VOI Obtained with the Three Models

Figure 4 presents the segmentation results in representative cases from the testing data of the JC dataset. The VOI predicted by the BraTS model shows a larger area that includes normal brain tissue, not only the brain tumor. However, the fine-tuning model specific to the facility in Case 3 only shows the area that appears to be the tumor.

Here, we present three cases with the ground truth and predicted VOI. The three cases are cases from both datasets whose segmentation task seemed to be either very easy or difficult for the BraTS model. The first case, Case 1, is BraTS19_TCIA08_242_1 from testing data of the BraTS dataset (Appendix A). Case 2 is from the testing data of the JC dataset (Appendix A). The segmentation of these two cases seemed to be an easy task for machine learning models; the Dice coefficient score in the BraTS model was high (0.969 and 0.926). The final case is Case 3 from the testing data of the JC dataset, which is presented in Figure 4. The BraTS model could not predict the VOI accurately; the Dice coefficient score was low (0.330). Six histograms are presented to inspect the difference between the difficult and easy segmentations.

In Figure 5A, we present the relationship between the four images. The GdT1 parts of the histograms of Case 1 and Case 2 have one peak around 1.8; however, Case 3 has no peak. Figure 5B shows the relationship between the T2 image and true VOI as a histogram. The VOI part of the histograms of Case 1 and Case 2 is located at the right edge (around 3) and occupies a relatively large portion. However, that of Case 3 is located centrally (around 2.5) and covers a small portion.

The tumor volume of Case 3 was small (12.9 mL). Then, further analysis was performed to study the correlation between the Dice coefficient and tumor volume. Appendix A is a scatter plot of the Dice coefficient score and tumor volume. When the correlation coefficient was calculated using Pearson product-moment correlation coefficient, there was no correlation between the Dice coefficient and tumor volume for all three model types. However, the correlation coefficient of the BraTS model was slightly higher than that of the JC model and the fine-tuning model (0.246 [BraTS model], 0.152 [JC model], and 0.156 [fine-tuning model]). In addition, when we focused on the cases with the Dice coefficient lower than 0.6, 60% (33/55) of the cases had a tumor volume of less than 50 mL. Judging from this fact, the trend that when the tumor volume is small (less than 50 mL), the Dice coefficient is also low (less than 0.6) was observed in all three model types.

## 4. Discussion

In the present study, we demonstrated that the significant performance degradation of a machine learning model for glioma segmentation was caused by the domain shift (Appendix A). Furthermore, we observed significant variation in the performance degradation among facilities. We believe that the fine-tuning method developed in this study would be a useful way to solve this issue (Figure 2 and Figure 4, and Appendix A).

Because of glioma’s invasive nature, it is difficult to determine the precise boundary between tumors and normal brain tissues in MR images. However, determining the precise boundary is considerably important for both high- and low-grade gliomas before the surgery because more extensive resection of the tumor is associated with better outcomes [8]. To prevent postoperative neurological dysfunction, preoperative recognition of neural anatomy and clear tumor boundary is needed.

In addition, it is also important to determine the boundary of gliomas accurately when evaluating the effect of the treatment in clinical trials based on response assessment criteria in neuro-oncology. Various clinical trials for gliomas are ongoing, and therapeutic effects are largely assessed by the radiological response of MR images. In multi-institutional clinical trials, the boundary must be created based on the same criteria, but it can differ among clinicians who evaluate it. Overall, the situation that we need to determine the precise boundary between a tumor and normal brain tissues in MR images has frequently occurred in clinical settings. Therefore, robust machine learning models for tumor segmentation are required.

The determination of the boundary between a tumor and normal brain tissues in gliomas relies on the abilities of skilled radiologists. For this reason, it is conceivable that a machine learning model for glioma segmentation is particularly useful for facilities where there are no skilled radiologists. The number of glioma cases in facilities where no skilled neuroradiologists are working is generally low. Given that numerous MR images are required to build a machine learning model for glioma segmentation, it would be unrealistic if a model has to be built from the beginning in each facility. Our fine-tuning method can offer the solution because it requires fewer than 20 glioma cases.

Although many researchers recognize that domain shifts reduce the performance of machine learning models, this problem has been poorly investigated [38,39,40]. One of the main reasons for poor investigation is the requirement of datasets from multiple facilities. In this regard, Pooch et al. recently reported that poor performance of models, due to the domain shift, was observed even in chest radiographs, which have less information and diversity among facilities than 3D brain volumes [41]. AlBadawy et al. also reported the poor performance of models caused by the domain shift in MR images of gliomas from two different facilities in The Cancer Imaging Archive (TCIA) public dataset [42]. This result is noteworthy because MR images of gliomas in the TCIA seem to be more homogeneous than those in the JC dataset. Homogeneity is an advantage when analyzing within the same dataset, but a disadvantage when extrapolating to other datasets.

Our JC dataset was collected from 10 facilities during an extremely long period of time from 1991 to 2015. We visualized the diversity of image resolutions of the JC dataset using the size of voxels (Appendix A). For example, the voxel size in the *z*-axis, which corresponds to the slice thickness, varied widely from 3.3 to 9 mm. MR scanners and image settings also varied. This fact indicates that the JC dataset is heterogeneous and likely represents real-world data.

We used the JC dataset and found highly frequent and severe performance degradation of a machine learning model caused by the domain shift. It should be noted that the severity of performance degradation varied substantially among domains (Figure 2 and Appendix A). This result implies that it is difficult to guarantee the performance of a machine learning model, which is trained only on a public dataset, such as the BraTS dataset.

Although the network architecture of U-Net is frequently used when building machine learning models for segmentation, it has rarely been investigated for fine-tuning. Amiri et al. recently assessed the effects of fine-tuning different layers of the U-Net model that was trained on 1000 segmented natural images for segmentation of breast ultrasound images [39], and concluded that setting the shallow layer as a learnable layer was more effective than setting the deep layer. They also mentioned that concordant results were not observed in the segmentation of radiographs.

In this study, we demonstrated that the most effective fine-tuning model is the down2_up2 model (learnable layers were those in 1st down, 2nd down, 1st_up, and 2nd_down blocks, Appendix A). The architecture of U-Net is divided into two phases; the first phase performs a convolution to obtain a latent representation, and the other phase reconstructs an image from the latent representation. The phase of obtaining a latent representation of U-Net is almost the same as that of a classic convolutional neural network (CNN). The CNN representations of shallow layers are lower-level features (e.g., lines, edges, and blobs), and the CNN representations of deep layers are more semantic and higher-level features. In our experience, the image differences among facilities seem to be caused by the diversity of instruments and methods in each facility. We assume that the image differences among facilities could be processed by low-level features, corresponding to shallow layers in CNN. Hence, it is reasonable to make layer groups that consist of a shallow layer block and a deep layer block, which is directly connected to the shallow layer. Our results are compatible with the results obtained by Amiri et al. [39].

As shown in Figure 5A, the shapes of the histograms of Case 1 and Case 2 are similar; however, that of Case 3 is slightly different. The differences in image quality cannot be normalized even after the Z-score conversion might have caused this histogram difference. Therefore, in Case 1 and Case 2, the VOIs account for a large percentage of the histogram and are located at the right tail. In other words, the tumors of Case 1 and Case 2 are large, and there are a few voxels having a higher voxel value than that of the part of the tumor. Segmentation of such cases is considered to be an easy task for the machine learning model. In Case 3, the region occupied by the VOI is small, and the VOI exists close to the center of the histogram. Segmentation of such cases appears to be difficult for the machine learning model. As shown in Figure 4, our fine-tuning method seems to work in Case 3 and could make the VOI close to the ground truth. Because a relatively small number of cases (fewer than 20 cases) are sufficient to show good performance, our method is useful when creating a machine learning model in a facility where there are a small number of glioma cases. Since we believe that all patients with gliomas should ideally be treated at a complex oncological facility, we hope that our machine learning method will help local hospitals, where such patients may visit for the first time, to appropriately direct patients to referral facilities.

As shown in Table 1, the data obtained from a single facility are always biased. To deploy a high-performance machine learning model, it is ideal to obtain unbiased data, such as the data from one large domain; however, this is rarely realized. Therefore, there may be two things that are required for building an accurate machine learning model in a clinical setting. First, it is important to collect as many images as possible from multiple facilities for building a model. Second, we must optimize the model corresponding to the ‘data bias’ of the facility. In our view, turning the two steps into a cycle can gradually create a ‘truly useful’ machine learning model in clinical sites.

Our study has some limitations. Firstly, our fine-tuning method works for our 3D U-Net model, but may not always work for other improved U-Net models. Furthermore, our fine-tuning method is likely to be useful in glioma segmentation; however, it is unclear whether it will work for other types of brain tumors, including brain metastases or primary central nervous system lymphoma. Further verification may clarify this point. The JC dataset is very large and heterogeneous as a glioma dataset; nonetheless, verification with another large dataset is needed in the future.

## 5. Conclusions

Our study explored the performance degradation of the machine learning model for glioma segmentation caused by the domain shift; the fine-tuning method developed in this study can be a useful way to solve this issue. Because our fine-tuning method only requires fewer than 20 cases, this method is useful when building a machine learning model for segmentation in a facility where the number of glioma cases is small.

## Figures and Tables

**Figure 1 cancers-13-01415-f001:**
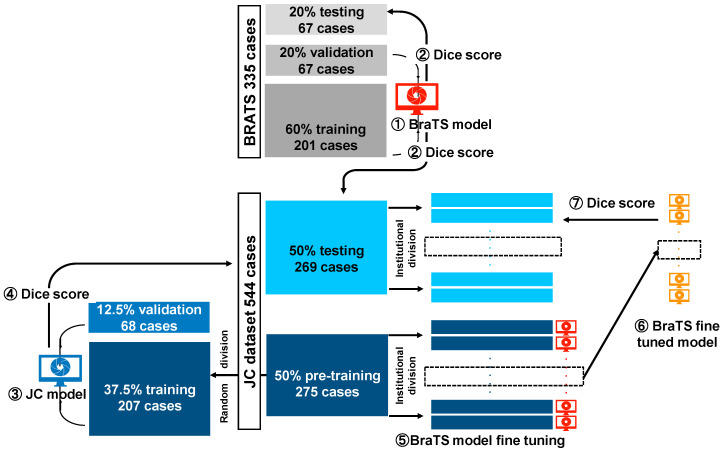
The overall flow of this research. Three types of machine learning models were built for segmentation: The Multimodal Brain Tumor Image Segmentation Benchmark (BraTS) model, the Japanese cohort (JC) model, and the fine-tuning model. (1) The BraTS model was trained in the training part of the BraTS dataset. (2) The BraTS model was evaluated by the Dice coefficient scores of the BraTS data set and the test portion of the JC data set. (3) The JC model was trained on the training portion of the JC data set. (4) The JC model was evaluated by the Dice coefficient score of the test portion of the JC data set. (5) The BraTS model was fine-tuned for optimal analysis at each facility. (6) The BraTS model fine-tuned by the aforementioned method, was named the fine-tuning model. (7) The fine-tuning model was evaluated with the Dice coefficient score of the test portion of the JC data set.

**Figure 2 cancers-13-01415-f002:**
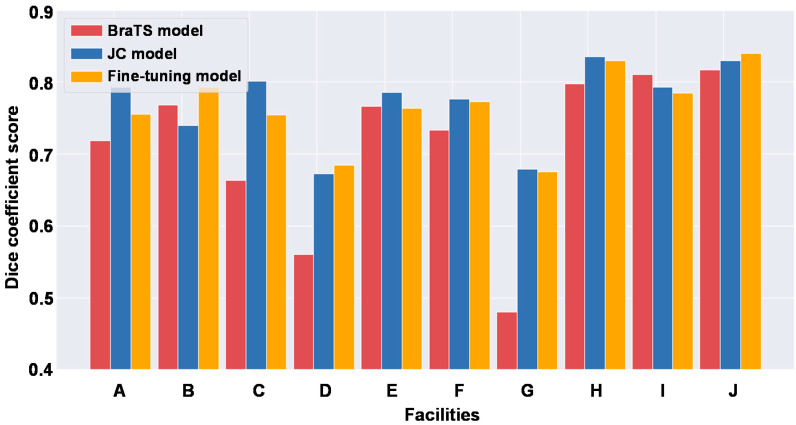
The results of three types of machine learning models for segmentation. A bar graph shows the Dice coefficient score for each facility. The horizontal axis is the facility, the vertical axis is the Dice coefficient score, and the colors indicate the types of machine learning models for segmentation. The Dice coefficient score of the fine-tuning model was significantly improved compared to that of the BraTS model and was comparable to that of the JC model.

**Figure 3 cancers-13-01415-f003:**
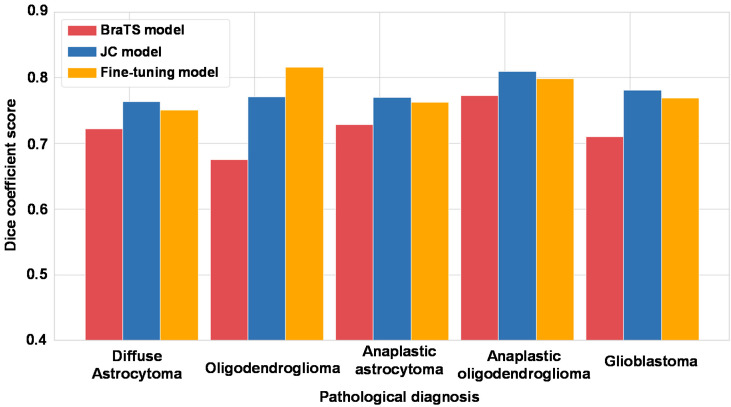
The results of three types of machine learning models for segmentation focused on pathological diagnosis. A bar graph shows the Dice coefficient score for each pathological diagnosis. The horizontal axis shows the pathological diagnosis, the vertical axis shows the Dice coefficient score, and the colors show the type of machine learning models for segmentation.

**Figure 4 cancers-13-01415-f004:**
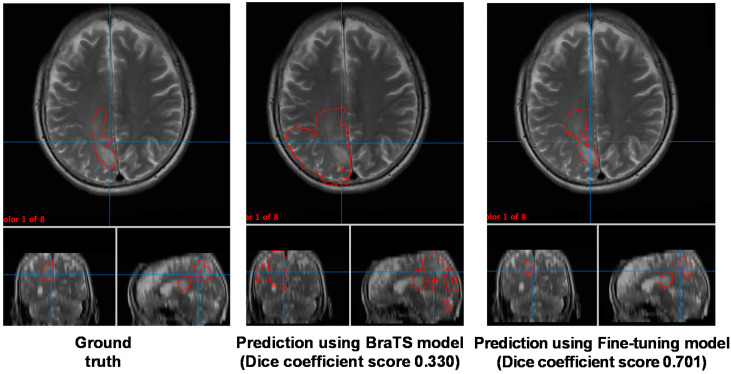
Segmentation results on Case 3. The left column is ground truth (made by a skilled radiologist), the middle is predicted by the BraTS model, and the right is predicted by fine-tuning model specialized for the facility has Case 3. The volume of interest (VOI) predicted by the BraTS model had an area that didn’t seem to be a brain tumor. However, that by fine-tuning model specialized for the facility has Case 3 only have an area that seemed to be a tumor.

**Figure 5 cancers-13-01415-f005:**
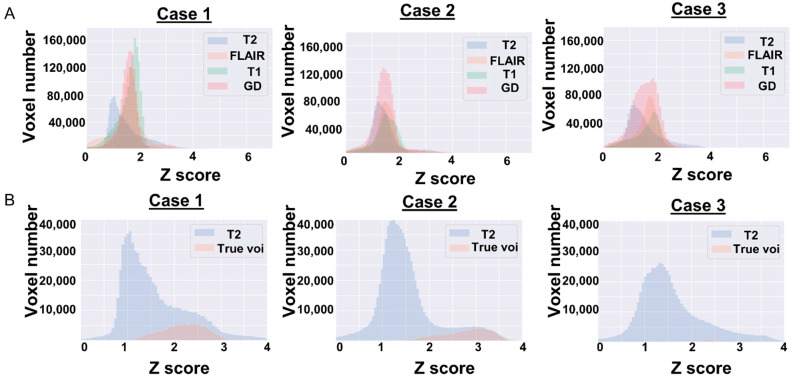
The results of comparison of image types in Case 1 from the BraTS dataset and Case 2 and Case 3 cases from the JC dataset are shown in histograms. The horizontal axis of the histogram represents the voxel value converted into a Z-score, and the vertical axis represents the number of voxels. (**A**) Each color represents image types. The GdT1 part of histograms of Case 1 and Case 2 have one peak around 1.8, however that of Case 3 has no peak. (**B**) The image histograms show the relationship between the T2 image (equivalent to the slice corresponding to the VOI) and true VOI. Blue histogram represents the T2 image, and orange represents true VOI. The VOI part of histograms of Case 1 and Case 2, located right edge (around 3), occupied a relatively large portion. But that of Case 3 was located central (around 2.5) and had a small portion.

**Table 1 cancers-13-01415-t001:** Clinical characteristics of the cases in this study.

Parameter	All Dataset	Facility A	Facility B	Facility C	Facility D	Facility E	Facility F	Facility G	Facility H	Facility I	Facility J
**Median age** **(range)**	60 (86–17)	54 (81–28)	64.5 (84–26)	64.5 (85–25)	66 (85–51)	59 (80–25)	57 (86–17)	60 (79–22)	55.5 (80–21)	54 (76–28)	61 (81–19)
Sex											
Male	293	92	20	50	8	32	17	21	23	16	23
Female	251	65	20	44	5	27	14	21	21	13	12
**LrGG or GBM**											
LrGG	218	71	18	0	0	31	18	25	23	18	14
GBM	326	86	22	94	13	28	13	17	21	11	21
**WHO grade**											
II	91	34	12	0	0	7	5	10	10	7	6
III	127	37	6	0	0	24	13	15	13	11	8
IV	326	86	22	94	13	28	13	17	21	11	21
**Pathological diagnosis**											
Diffuse astrocytoma	66	27	9	0	0	4	3	9	7	4	3
Anaplastic astrocytoma	88	25	3	0	0	16	6	14	10	7	7
Oligodendroglioma	25	7	3	0	0	3	2	1	3	3	3
Anaplastic oligodendroglioma	39	12	3	0	0	8	7	1	3	4	1
Glioblastoma	326	86	22	94	13	28	13	17	21	11	21

LrGG = lower grade glioma, GBM = glioblastoma.

## Data Availability

The data presented in this study are available in this article (and Appendix A).

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
