# Peer review of "Fine-Tuning Approach for Segmentation of Gliomas in Brain Magnetic Resonance Images with a Machine Learning Method to Normalize Image Differences among Facilities"

_cancers, 2021, doi:10.3390/cancers13061415_

Round 1
Reviewer 1 Report
Image segmentation in digital image processing and computer analysis of radiological information has multiple practical applications in medicine. In neuro-oncological surgery the typical application is the localization of tumors, measurement of tissue volume, virtual simulation, surgical pre op and intra-op planning, evaluation of radicality and long term monitoring. Diffuse gliomas are the most common primary brain tumors. The general prognosis of tumors grade II-IV is unfavorable with various life expectancies despite of combined oncological treatment. The radical „radiological„ resection is the one of the most important factors influencing prognosis and survival of these patients. Proper segmentation and appropriate assessment of tumor extension has lead to better treatment planning and subsequent execution of safe radical radiological resection. Distant tumor infiltration, typical for diffuse gliomas, excludes the possibility of segmentation of absolute tumor margins. Intraoperative application is still limited by existence of gradual brain shift. One of the most important benefits of segmentation is long term monitoring of residual tumor or its recurrence. Manual segmentation is an extremely boring and time-consuming procedure.
Authors present a fine-tuning approach for segmentation with a machine learning method specific for different facilities. Three models of segmentation were compared based on four basic MRI sequences from 544 subjects of a Japanese cohort dataset. The MRI data were acquired from ten different facilities. All three models were compared using the Dice coefficient score. One of the most important results of this study is the evaluation of the domain shift and machine learning models for different facilities. The authors declare a small number of cases (fewer than 20 cases) to get sufficient results. The results from a single facility were biased.
Precise automatic segmentation together with the other different AI modalities and their clinical application will probably partially improve the safety of neurosurgical therapy of brain tumors, subsequent oncological treatment and general outcome of these patients. More data from different research groups are necessary to implant these methods to routine daily practice. Future evaluation of automatic segmentation accuracy of radiological margins of diffuse gliomas together with clinical correlation and actual histological borders is also necessary.
Comments:
The data of diffuse gliomas were collected from 10 facilities during an extremely long period of time from 1991 to 2015. The quality of MR throughout 25 years is most likely variable. Describe or exclude the differences in the results.
The group of diffuse gliomas (grade II, III, IV) are radiologically extremely different (contrast agent enhancement, role of edema). Describe the difference between the groups.
An experienced oncological neurosurgeon is still better than a skilled radiologist in my personal opinion. The gliomas should not be treated in low volume centers, but in complex oncological facilities.
The Dice coefficient score should be described for as readers do not have experience with the theory of image segmentation.
I don’t understand the benefit of definition VOI including edema? The peritumoral functional edematic brain tissue should be preserved.
Did you try to verify the borders of tumor histologically during subsequent resection?
The paper was presented and published as an abstract NIMG-29 in Neuro-oncology in November 2020. This fact should be mentioned.
Some typing errors should be corrected (for example line 329 enema).
Literature:
Liu, Zhongqiang. (2020). Automatic Segmentation of Non-Tumor Tissues in Glioma MR Brain Images Using Deformable Registration with Partial Convolutional Networks.
Jean, Stawiaski. (2017) A Multiscale Patch Based Convolutional Network for Brain Tumor Segmentation. CoRR abs/1710.02316
Author Response
Dear Professor,
We are grateful for your consideration of our manuscript entitled “Fine-tuning Approach for Segmentation of Gliomas in Brain Magnetic Resonance Images with a Machine Learning Method to Normalize Image Differences among Facilities” (manuscript ID: cancers-1071838) by Takahashi et al. and appreciate your helpful comments.
Replies to the comments are as follows:
Comment 1: The data of diffuse gliomas were collected from 10 facilities during an extremely long period of time from 1991 to 2015. The quality of MR throughout 25 years is most likely variable. Describe or exclude the differences in the results.
Reply: Thank you for your important comment. We certainly agree with your statement that MRI data collected from 10 institutions over 15 years contains a wide range of quality. Therefore, we visualized the diversity of image resolution and added a new figure and a table in supplementary data (Supplementary Figure S12 and Table S4). Also, we added the following description in the Discussion section of the revised manuscript (lines 373-378).
“Our JC dataset was collected from 10 facilities during an extremely long period of time from 1991 to 2015. We visualized the diversity of image resolutions of the JC dataset using size of voxels (Supplementary Figure S12 and Table S4). For example, the voxel size in the z-axis, which corresponds to the slice thickness, varied widely from 3.3 to 9 mm. MR scanners and image settings also varied. This fact indicates that the JC dataset is heterogeneous and likely represents real-world data.”
Comment 2: The group of diffuse gliomas (grade II, III, IV) are radiologically extremely different (contrast agent enhancement, role of edema). Describe the difference between the groups.
Reply: Thank you for your important comment. As you pointed out, we also recognize the radiological differences among diffuse gliomas (WHO grade II, III and IV). We added the following description in the Results section of the revised manuscript (line 303-306), and we performed further analysis and added new figure and table for better understanding (Figure 3 and Supplementary Table S3).
“Then, we focused on pathological diagnosis and comparison of three models. We show the results in Figure 3 and Supplementary Table S3. The Dice coefficient scores of JC model and Fine-tuning models tended to be better than those of the BraTS model, especially for oligodendroglioma and glioblastoma.”
Comment 3: An experienced oncological neurosurgeon is still better than a skilled radiologist in my personal opinion. The gliomas should not be treated in low volume centers, but in complex oncological facilities.
Reply: Thank you for your important comment. We agree with your opinion and all the patients with gliomas should ideally be treated in a large comprehensive center. However, it is also true that patients with glioma can visit local hospital where expert neurosurgeon or radiologist does not work, and we believe our machine learning method will help the local hospitals to guide the patients appropriately to the center hospitals. We amended the manuscript by adding a description in the Discussion section of the revised manuscript (lines 414 - 417).
“Since we believe that all patients with gliomas should ideally be treated at a complex oncological facility, we hope that our machine learning method will help local hospitals, where such patients may visit for the first time, to appropriately direct patients to referral facilities.”
Comment 4: The Dice coefficient score should be described for as readers do not have experience with the theory of image segmentation.
Reply: Thank you for your important comment. In response to your suggestion, we have created a new section in Materials and Methods of the revised manuscript that explains the Dice coefficient score as follows (lines 245 - 250).
“2.9. Performance Evaluation of Segmentation Models
The performance of the segmentation models was evaluated by Dice coefficient scores. The Dice coefficient is a score that indicates the similarity between two samples. In the case of two images, the Dice coefficient score was calculated by dividing the number of pixels in the overlapping area times two by the number of pixels in both images. Therefore, the Dice coefficient score ranges zero to one, and is one if the two images are an exact match.”.
Comment 5: I don’t understand the benefit of definition VOI including edema? The peritumoral functional edematic brain tissue should be preserved.
Reply: Thank you for your important comment. Pathologically, perifocal edema surrounding tumors is known to contain invasive tumor cells, and MRI may contain some signals caused by non-enhancing tumor lesions in these areas. Since this is an important point, we have added this discussion in the Materials and Methods section of the revised manuscript (lines 171-173).
“We included the T2 hyper lesions in the VOI because it may contain non-enhancing tumor lesions. We think that the advantages of including non-enhancing tumor outweigh than the disadvantages of including ‘pure edema’ that may make noise.”
Comment 6: Did you try to verify the borders of tumor histologically during subsequent resection?
Reply: Thank you so much for your keen comment. Unfortunately, we have not collected subsequent information including resection, but this is a very important aspect for future research.
Comment 7: The paper was presented and published as an abstract NIMG-29 in Neuro-oncology in November 2020. This fact should be mentioned.
Reply: Thank you for your comment. As you pointed out, some part of this study was actually presented at the 25th annual meeting of Society for Neuro-Oncology. We added a following description next to Conflicts of Interest at the end of the revised manuscript (lines 466 - 467).
“Meeting presentation: Some part of this study was presented at 25th Society for Neuro-Oncology meeting as an abstract number of NIMG-29.”
Comment 8: Some typing errors should be corrected (for example line 329 enema).”
Reply: Thank you for pointing out our typing errors. We carefully checked and revised the manuscript.
Comment 9: Literature:
Liu, Zhongqiang. (2020). Automatic Segmentation of Non-Tumor Tissues in Glioma MR Brain Images Using Deformable Registration with Partial Convolutional Networks.
Jean, Stawiaski. (2017) A Multiscale Patch Based Convolutional Network for Brain Tumor Segmentation. CoRR abs/1710.02316”
Reply: We appreciate your guidance on these articles. We included these two papers into references (ref number 14 and 15).
Thank you so much for sharing your precious time with us. Our manuscript was significantly improved based on the constructive comments.

Reviewer 2 Report
In this study, the authors evaluated the performance degradation of machine learning models for segmentation of gliomas in brain magnetic resonance images caused by domain shift and proposed possible solutions. They demonstrated that performance degradation can be improved by fine-tuning methods with small number of cases from different facilities. They developed a facility-specific machine learning model from small dataset. They concluded that their fine-tuning method could be a practical solution to overcome performance degradation of small dataset.
This is an interessting and relevant paper about an important topic: introduction and establishment of ML methods for rare diseses and small data set. However, the manuscript has a somewhat confused structure and some methods should be described in more detail. Therefore I recommend aceptance after revision.
Comments:
The first, second and third paragraph of the Results section (line 115 - 144) describe methods and should be transferred into the Methods section. The authors should include an explanation of the meaning of the numbers 1 to 7 in Figure 1 into the text or the figure legend.
A general problem with the manuscript is that the methods are described in three different sections of the manuscript. This is not ideal for a technical paper.
The MRI data were collected from 951 patients (line 307), but the authors used images from “544 of the 673 subjects“. The authors should explain pateint selection and exclusion criteria.
Lines 315 - 317: The authos should provide ranges for the most important geometric sequence parameters used, i.e.number of slices, slice thickness, resolution. This is an important information about the heterogeneity of the data used in the study.
Were MR sanners from different vendors used?
Line 342: The authors should discribe the “three types of machine learning models” in more detail here.
How was the Dice coefficient score calculated. What is its meaning.
Line 355: The authors should - at least briefly - describe the architecture of their machine learning model. The figure caption of Figure S2 is not very informative.
Sttatistical analysis: Was a correction for multiple comparisons performed?
Supplementary Method:
Lines 19 - 21, 28 - 34: These are results and should be transfered into the Results section.
Results:
Figure 3 shows an interessting analysis. The small portion in the right edge of the histogram of case 3 (Fig. 3b) might be associated with the much smaller lesion volume of this case comapered to cases 1 and 2. I wondering if there is a correlation between segmentation quality and lesion volume? In other words, the smaller the lesion volume the worse the segmentation result. The authors should included an analysis addressing this issue.
Figure 4 should be included before Figure 3.
Author Response
Dear Professor,
We are grateful for your consideration of our manuscript entitled “Fine-tuning Approach for Segmentation of Gliomas in Brain Magnetic Resonance Images with a Machine Learning Method to Normalize Image Differences among Facilities” (manuscript ID: cancers-1071838) by Takahashi et al. and appreciate your helpful comments.
Replies to the comments are as follows:
Comments 1 and 3: The first, second and third paragraph of the Results section (line 115 - 144) describe methods and should be transferred into the Methods section.”
“A general problem with the manuscript is that the methods are described in three different sections of the manuscript. This is not ideal for a technical paper.”
Reply: Thank you for your important comment. We agree with your idea that materials and methods should be described in the appropriate place, and have boldly changed the first, second, and third paragraphs of the Results section, which contains Table 1 and Figure 1, to the Materials and Methods section as you recommended. Also, we moved Materials and Methods between Introduction and Results for better understanding by the reader. We have also renamed the sections ("2.2. JC Dataset" and "2.3. BraTS Dataset") for better understanding by the reader, and the details of the cases and MRI sequences used in this study are described here, integrating the previous repeated descriptions.
Comment 2: The authors should include an explanation of the meaning of the numbers 1 to 7 in Figure 1 into the text or the figure legend.
Reply: Thank you for your important comment. As you suggested, I have included a detailed explanation of the numbers 1-7 in the legend of Figure 1.
Comment 3: The MRI data were collected from 951 patients (line 307), but the authors used images from “544 of the 673 subjects“. The authors should explain patient selection and exclusion criteria.
Reply: Thank you for your comment, and we apologize for any confusion this may have caused you due to our inadequate explanation. To explain precisely, since the clinical and genetic data of 951 patients were reported in the previous study, we first tried to collect MR images from these 951 patients. However, there were 278 cases in which appropriate preoperative MR images were not digitized (mostly older cases, with MR images on film or printed), and 673 cases in which at least one sequence had appropriately digitized preoperative MR images. As a result, 544 cases were included in this study. We have modified the description (line 140) and added the following description to the Materials and Methods section of the revised manuscript (lines 144-145).
“Of the 951 cases, 673 cases had preoperative digital MR images in at least one sequence, and 544 cases met our criteria described below.”
Comment 4: Lines 315 - 317: The authors should provide ranges for the most important geometric sequence parameters used, i.e. number of slices, slice thickness, resolution. This is an important information about the heterogeneity of the data used in the study.”
Reply: Thank you for your important comment. We agree with your idea that the sequence parameters are important. Although not all parameters can be displayed immediately, to illustrate the heterogeneity of the data used in this study, we have added the following statement in the Discussion section of the revised manuscript (lines 373-378) and added a new supplemental figure (Figure S12) and a table (Table S4) describing the range of voxel sizes and slice thicknesses in the MRI data.
“Our JC dataset was collected from 10 facilities during an extremely long period of time from 1991 to 2015. We visualized the diversity of image resolution of JC dataset using size of voxels (Supplementary Figure S12 and Table S4). For instance, z axis voxel sizes, corresponding to the slice thickness, ranged 3.3 to 9 mm largely varied. MR scanners and image settings also varied. This fact indicates the JC dataset is heterogeneous and is likely to represent real-world data.”
Comment 5: Were MR scanners from different vendors used?
Reply: Thank you for your comment. Yes, the MR images are made by scanners from various venders. We included this point in the above description in Comment 4 (line 376 to 377).
Comment 6 Line 342: The authors should describe the “three types of machine learning models” in more detail here.
Reply: Thank you for your valuable comment, and we agree to describe the three models in more detail. We have added the following explanation to the Materials and Methods section of the revised manuscript (lines 200-206). Also, we have included the information in Supplementary Table S1.
“As shown in Supplementary Figure S2, we grouped all the layers between the two element-wise sums or merged operation layers into one block for further technical convenience. Most of the blocks consist of three 3D convolution layers and one dropout layer. The concept of each group corresponds to the blue shaded areas in Supplementary Figure S2. Each block was named in order from the shallowest to the deepest block as ‘1st down’, ’2nd down’, …‘5th down’ and then from the deepest to the shallowest block as ’1st up’, ’2nd up’,… ’5th up’. The optimizer was set to AdaGrad, and the learning rate was set to 0.005.”
Comment 7: How was the Dice coefficient score calculated. What is its meaning.
Reply: Thank you for your suggestion. To explain the Dice coefficient score, the following description was added in the Materials and Methods section of the revised manuscript (lines 245-250).
“2.9 Performance Evaluation of Segmentation Models
The performance of segmentation models was evaluated by Dice coefficient scores. Dice coefficient score gauge the similarity of two samples. In case of gauging two images, the Dice coefficient score was calculated by the double number of pixels in the area of overlap dividing by the number of pixels in both images. Therefore, Dice coefficient score ranges zero to one, and if the two images are an exact match, it is one.”
Comment 8: Line 355: The authors should - at least briefly - describe the architecture of their machine learning model. The figure caption of Figure S2 is not very informative.
Reply: Thank you for your important suggestion. We have added the following description to the legend in Supplementary Figure S2, as well as Table S4 and a new reference (Ref. 34) in the revised manuscript.
“All layers between two element-wise sums or merged operation layers into one block was grouped. Most blocks contain three 3D convolution layers and one dropout layer. The concept of each group corresponds to the blue shaded areas in Figure S2. Each block was named in order from the shallowest to the deepest block as ‘1st down’, ’2nd down’, …‘5th down’ and then from the deepest to the shallowest block as ’1st up’, ’2nd up’,… ’5th up’. The optimizer was set to AdaGrad, and the learning rate was set to 0.005..”
Comment 9: Statistical analysis: Was a correction for multiple comparisons performed?”
Reply: Thank you for your comment. We performed the test using the Games-Howell Post-Hoc Test, which is a method of multiple comparisons similar to the Bonferroni correction.  In addition, Welch's ANOVA is used as analysis of variance and Games-Howell Post-Hoc Test is used as post-test. Welch's ANOVA and Games-Howell Post-Hoc Test are suitable for this setting because they can be used in environments with heterogeneous variances and sample sizes.
Comment 10 Supplementary Method: Lines 19 - 21, 28 - 34: These are results and should be transferred into the Results section.”
Reply: Thank you for your comment and we agree with the your recommendation. We created a new section (2.7. Finding the Best Fine-Tuning Method) in the Materials and Methods section of the revised manuscript, and transferred the most of description in Supplementary Method into the new section (lines 211-231).
Comment 11. Results: Figure 3 shows an interesting analysis. The small portion in the right edge of the histogram of case 3 (Fig. 3b) might be associated with the much smaller lesion volume of this case compared to cases 1 and 2. I wondering if there is a correlation between segmentation quality and lesion volume? In other words, the smaller the lesion volume the worse the segmentation result. The authors should include an analysis addressing this issue.
Reply: Thank you so much for your interest in our analysis. As you pointed out, correlation between segmentation quality and lesion volume is an important point, so we performed further analysis. Then, we have added the following description (lines 326-335) and a new supplementary figure (Figure S10).
“The tumor volume of case 3 was small (12.9 ml). Then, further analysis was performed to study the correlation between Dice coefficient and tumor volume. Supplementary Figure S10 is a scatter plot of Dice coefficient score and tumor volume. When the correlation coefficient was calculated using Pearson product-moment correlation coefficient, there was no correlation between Dice coefficient and tumor volume for all three model types. However, the correlation coefficient of the BraTS model was slightly higher than that of the JC model and the fine-tuning model (0.246 [BraTS model], 0.152 [JC model], and 0.156 [Fin-tuning model]). In addition, when we focused on the cases with the Dice coefficient lower than 0.6, 60% (33/55) of the cases had a tumor volume of less than 50 ml. Judging from this fact, the trend that when the tumor volume is small (less than 50 ml), the Dice coefficient is also low (less than 0.6) was observed in all three model types..
Comment 12. Figures: Figure 4 should be included before Figure 3.
Reply: Thank you for your comment. In response to your suggestion, we have changed the order of Figure 4 and Figure 3.
Thank you so much for sharing your precious time with us. Our manuscript was significantly improved based on the constructive comments.

Round 2
Reviewer 2 Report
All of my questions were adequately answered. Congratulations to the authors for the excellent paper.